# Challenges of Nontherapeutic Laparotomy in Patients with Peritoneal Surface Malignancies Selected for Cytoreductive Surgery and Hyperthermic Intraperitoneal Chemotherapy

**DOI:** 10.3390/cancers17091445

**Published:** 2025-04-25

**Authors:** Elena Gil-Gómez, Alida González-Gil, Vicente Olivares-Ripoll, Álvaro Cerezuela-Fernández de Palencia, Francisco López-Hernández, Álvaro Martínez-Espí, Jerónimo Martínez-García, Francisco Barceló, Alberto Rafael Guijarro-Campillo, Pedro Antonio Cascales-Campos

**Affiliations:** 1Peritoneal Carcinomatosis and Sarcomas Unit, Department of Surgery, Hospital Universitario Virgen De La Arrixaca, IMIB-Arrixaca, 30120 Murcia, Spain; elenagilgomez@hotmail.com (E.G.-G.); alidagonzalezgil@gmail.com (A.G.-G.); vicenteolivaresripoll@gmail.com (V.O.-R.); alvaro.cerezuela@gmail.com (Á.C.-F.d.P.); pacolopezh3@gmail.com (F.L.-H.); alvaromartinez997@gmail.com (Á.M.-E.); 2Department of Medical Oncology, Hospital Universitario Virgen De La Arrixaca, IMIB-Arrixaca, 30120 Murcia, Spain; jeronimo@seom.org; 3Department of Gynecologic Oncology, Hospital Universitario Virgen De La Arrixaca, IMIB-Arrixaca, 30120 Murcia, Spain; fjbarcelovalcarcel@hotmail.com (F.B.); argc777@gmail.com (A.R.G.-C.); 4Department of Surgery, University of Murcia, 30100 Murcia, Spain

**Keywords:** peritoneal carcinomatosis, peritoneal metastases, intraperitoneal, chemotherapy, cytoreductive surgery, open and close

## Abstract

This study analyzed outcomes in patients with peritoneal surface malignancies who were initially considered to be candidates for cytoreductive surgery (CRS) with hyperthermic intraperitoneal chemotherapy (HIPEC) but were found to have unresectable disease during exploratory laparotomy. Among 486 patients evaluated between 2008 and 2022, 46 (9.4%) procedures were aborted due to extensive disease, especially with massive small bowel involvement. The median surgery duration was 90 min. Postoperative complications occurred in 22% of cases, and the mortality rate was 4.3%. Survival was significantly lower in patients who did not receive palliative systemic chemotherapy (4 months vs. 15 months). The findings highlight that nontherapeutic exploratory laparotomies carry considerable risks. Therefore, improving preoperative staging with advanced technologies such as radiomics and laparoscopy is essential to reduce the number of patients undergoing unnecessary surgery when the disease is ultimately found to be unresectable.

## 1. Introduction

In carefully selected patients, peritoneal surface malignancies can be cured [1]. Over the past three decades, a paradigm shift has redefined peritoneal carcinomatosis as a locoregional stage of peritoneal surface malignancies [2]. This condition can be managed through cytoreductive surgery (CRS) with peritonectomy procedures to address macroscopic disease, followed by hyperthermic intraperitoneal chemotherapy (HIPEC) to treat the microscopic component responsible for recurrence [3]. This treatment approach requires a coordinated multidisciplinary framework and must be performed at specialized centers with sufficient expertise due to the associated steep learning curve [4].

The combination of CRS and HIPEC has significantly improved the prognosis of various peritoneal surface malignancies, becoming the standard of care for conditions such as pseudomyxoma peritonei and malignant peritoneal mesothelioma [5,6]. In ovarian cancer, the randomized control trial OVHIPEC-1 demonstrated that HIPEC with cisplatin after interval CRS and neoadjuvant chemotherapy improved both disease-free and overall survival without significantly increasing morbidity, mortality, or negatively affecting quality of life [7]. Additional studies, including those by Cascales-Campos et al., Lim et al., and metanalysis published by Filis et al., have confirmed similar benefits in comparable clinical settings, leading to the inclusion of HIPEC in the National Comprehensive Cancer Network guidelines for ovarian cancer [8,9,10]. Similarly, the CHIPOR trial showed improved survival rates in platinum-sensitive recurrences treated with HIPEC, further supporting its role in selected cases [11]. In colorectal peritoneal carcinomatosis, however, the PRODIGE 7 trial found no added survival benefit with HIPEC using oxaliplatin. Nonetheless, median survival in both arms exceeded 40 months, far surpassing outcomes seen with systemic chemotherapy alone, highlighting the fundamental role of CRS in these patients [12]. Emerging evidence, such as the HIPECT4 trial, suggests that mitomycin-C-based HIPEC may enhance locoregional control in high-risk cases of T4 colorectal cancer [13,14,15,16]. On the other hand, prophylactic HIPEC in colorectal cancer, as investigated in the PROPHYLOCHIP and COLOPEC trials, has not demonstrated significant benefits, highlighting the need for further research in this area.

CRS and HIPEC are associated with procedure-related morbidity and mortality. However, in experienced groups, postoperative outcomes are comparable to those of other major surgical oncological procedures [17]. Sometimes, CRS must be aborted when disease is deemed unresectable during surgical exploration, despite preoperative evaluations suggesting a favorable scenario for CRS. These “open and close” procedures can generate complications without any benefit, particularly in patients with fragile profiles due to previous systemic treatments and their performance status related to the disease.

The primary objective of this study was to analyze morbidity, mortality, and survival outcomes in patients with peritoneal surface malignancies who were initially candidates for CRS with HIPEC but were found to have unresectable disease, leading to nontherapeutic exploratory laparotomy. We also discuss strategies to minimize the incidence of “open and close” procedures, including the potential role of laparoscopy in enhancing patient selection.

## 2. Materials and Methods

The experience of our referral center for treating peritoneal surface malignancies was evaluated from January 2008 to December 2022. All patients underwent rigorous selection, ensuring adequate cardiac, renal, hepatic, and bone marrow function, as well as an American Society of Anesthesiologists (ASA) score suitable for surgery. Patients classified as ASA IV or with poor performance status (ECOG > 2) were excluded. Surgery was performed with the intent of achieving optimal cytoreduction to no visible residual disease. Patients were excluded if they experienced disease progression during preoperative systemic chemotherapy treatment or if disease was found to be present outside the peritoneal cavity. This single-center retrospective study used a prospective database designed at the beginning of our program for the treatment of peritoneal surface malignancies. Follow-up concluded in January 2024.

In our center, laparoscopy was only performed in patients with uncertain resectability, exclusively as a “one-step” approach. If laparoscopic exploration confirmed resectability, CRS and HIPEC were performed during the same procedure using an open approach through a laparotomy. Our protocol does not include laparoscopy as a separate preliminary step prior to definitive CRS and HIPEC. Patients with ovarian cancer who underwent staging laparoscopy for biopsy before neoadjuvant chemotherapy were excluded. Preoperative imaging studies were reviewed with an experienced radiologist to assess eligibility, and all patients provided informed consent. After nontherapeutic laparotomy, which included biopsies of the disease in addition to laparotomy, and subsequent evaluation of the peritoneal cavity, adverse events were classified using the National Cancer Institute Common Terminology Criteria for Adverse Events (CTCAE version 4.0). Minor complications were defined as those requiring no treatment (grade I) or medical management alone (grade II). Severe complications required invasive procedures (grade III), reoperation or ICU admission (grade IV), or resulted in death (grade V). All complications were assessed within 90 days of surgery.

A descriptive analysis was performed, presenting data as medians (range), means (standard deviation), or frequencies (and percentages). Overall survival was calculated from the date of surgery using Kaplan–Meier analysis (SPSS Statistics for Windows, Version 25.0; IBM Corp., Armonk, NY, USA).

## 3. Results

Between January 2008 and December 2022, 489 patients with peritoneal surface malignancies were selected for CRS with HIPEC. Of these, 46 patients (9.4%) were found to have unresectable disease and underwent exploratory laparotomy with biopsies. The cohort included 43 women (93.5%) and 3 men (6.5%) with a median age of 62 years (range: 35–79). The most common indication was ovarian cancer (30 patients, 65%). Preoperative comorbidities were recorded in 24 patients, the most common being hypertension and diabetes. Primary CRS followed neoadjuvant systemic chemotherapy in 28 patients (61%), while the remaining 18 patients underwent surgery for recurrent disease. The main clinicopathological characteristics of the patients included in this study are shown in Table 1.

The median peritoneal cancer index (PCI) was 20 (range: 12–39). The primary reasons for unresectability were diffuse disease (28 patients) and massive involvement of the small intestine (13 patients). Median operative time was 90 min (range: 60–180). Postoperative complications occurred in 10 patients (22%), including grade II in 3 patients, grade III in 2, grade IV in 3, and grade V (mortality) in 2 patients (4.3%). Three patients required readmission, and one underwent repeat laparotomy for intestinal obstruction. Postoperative outcomes after nontherapeutic laparotomy in the patients in this series are included in Table 2.

Median overall survival was 11 months. Patients who received palliative systemic chemotherapy had significantly longer survival than those who did not (15 vs. 4 months; *p* < 0.01, Figure 1). Poor general condition or rapid disease progression was the reason for not indicating palliative systemic chemotherapy in four patients, while three refused treatments with systemic chemotherapy.

## 4. Discussion

In our experience with the treatment of peritoneal surface malignancies, fewer than 10% of patients selected for CRS with HIPEC had unresectable disease over a 14-year period. Among these, patients who did not receive palliative systemic chemotherapy after nontherapeutic laparotomy had particularly poor survival outcomes. These laparotomies with biopsies were not exempt from mortality rates, which reached 4%. The nontherapeutic laparotomy rate in our study is at the lower end of the range reported in other publications, which may exceed 20% to 25% of all patients selected for CRS plus HIPEC [18,19].

The most common reasons for performing an “open-and-close” procedure in the literature are diffuse small bowel involvement, the presence of a previous ostomy, and an American Society of Anesthesiologists (ASA) physical status of III. In our series, the most common cause of unresectable disease was diffuse disease massively affecting the small intestine, which had not been identified during preoperative imaging [20]. Advances in imaging technology offer promising alternatives to conventional contrast-enhanced computed tomography (CT) for the preoperative detection of peritoneal implants, especially small lesions that often go underestimated. Diffusion-weighted magnetic resonance imaging (DW-MRI) has emerged as a valuable tool for identifying peritoneal implants, providing superior sensitivity in detecting small nodules compared to traditional CT [21,22]. Furthermore, radiomics, a method that extracts quantitative features from medical images, enables the identification of subtle patterns associated with peritoneal disease that may not be visually apparent [23]. Additionally, artificial intelligence algorithms, including machine learning and deep learning, are being increasingly applied in preoperative evaluations. These AI-based approaches can enhance the detection of peritoneal implants by analyzing complex datasets and improving the accuracy of disease staging. Integrating these advanced imaging techniques into clinical practice could significantly reduce the need for diagnostic laparoscopies and laparotomies, offering a more precise and non-invasive assessment of peritoneal disease burden [23].

Diffuse small bowel involvement is a common cause of unresectability and can often only be determined through direct surgical visualization. Laparoscopy is a valuable tool for assessing small bowel involvement. Despite the potential risks associated with prior surgeries and the possibility of diffuse peritoneal cavity involvement, laparoscopy is considered a safe procedure [24]. However, it has limitations in accurately assessing peritoneal extension and calculating the peritoneal cancer index (PCI). The implementation of an initial laparoscopic evaluation protocol has been shown to reduce the rate of nontherapeutic laparotomies in some studies. For instance, Yurttas et al. reported a reduction in nontherapeutic laparotomies from 35% to 20%, although they noted that the laparoscopic PCI underestimated findings during laparotomy in up to 63% of cases [25]. Importantly, the laparoscopic procedure should always be performed by the same surgical team responsible for the subsequent CRS and HIPEC to ensure consistency and accuracy.

The need to abort CRS with HIPEC has significant implications for managing waiting lists and places a considerable financial burden on institutions. While these factors warrant attention, our group considers them secondary concerns that can be addressed through alternative management strategies focused on patient-centered care. A less explored aspect is the prognostic impact of nontherapeutic laparotomy in these patients. Surgical procedures carry inherent risks, particularly for patients with advanced cancer, who must temporarily discontinue systemic chemotherapy both pre- and postoperatively. In our series, 15% of patients experienced postoperative complications, and two patients (4%) died postoperatively. Additionally, seven patients (15%) were not treated with palliative systemic chemotherapy after surgery and were instead managed with best supportive care, resulting in markedly reduced overall survival of 4 months compared to 15 months in patients who were able to resume chemotherapy. These findings underscore the critical need to minimize unnecessary surgical interventions in this vulnerable population.

The negative impact of nontherapeutic laparotomy has also been studied in other cancer surgeries. Guo et al. reviewed nontherapeutic laparotomies in patients with cancers of various etiologies and reported findings like those in our study [26]. In a series of 345 patients with aborted cancer surgeries, the primary reason was unexpected tumor unresectability or the presence of occult metastatic disease. Approximately 40% of these patients did not receive postoperative systemic chemotherapy. Patients who resumed systemic therapy lived significantly longer than those who did not (18.6 vs. 5.4 months). A recent publication by the Trans-Atlantic Pancreatic Surgery Consortium, involving patients with locally advanced pancreatic cancer, revealed that postoperative mortality in patients who underwent nontherapeutic laparotomy was similar to that observed in patients who underwent major resection procedures for disease removal [27].

The primary limitation of this study is its retrospective design. Although the percentage of nontherapeutic laparotomies falls within the lower range reported in the literature, we question whether this percentage could be further reduced by incorporating systematic laparoscopy into our management protocol for these patients. Nevertheless, despite deciding not to include systematic laparoscopy in our approach, we believe that a single-stage strategy may be more suitable to avoid the need for a second, independent surgical procedure, although this opinion could be controversial. Another limitation of this study is the inclusion of a heterogeneous group of tumors, which may complicate the interpretation of prognostic outcomes.

## 5. Conclusions

Exploratory laparotomy in patients with peritoneal surface malignancies considered for CRS with HIPEC is not without complications. Furthermore, it may prevent patients from receiving palliative systemic chemotherapy due to associated morbidities and functional decline, which are linked to poor prognosis.

Improving preoperative staging with new technologies such as radiomics and laparoscopy is expected to decrease the number of patients undergoing nontherapeutic laparotomy, although this unfortunate scenario and its complications may not be entirely avoidable in daily clinical practice.

## Figures and Tables

**Figure 1 cancers-17-01445-f001:**
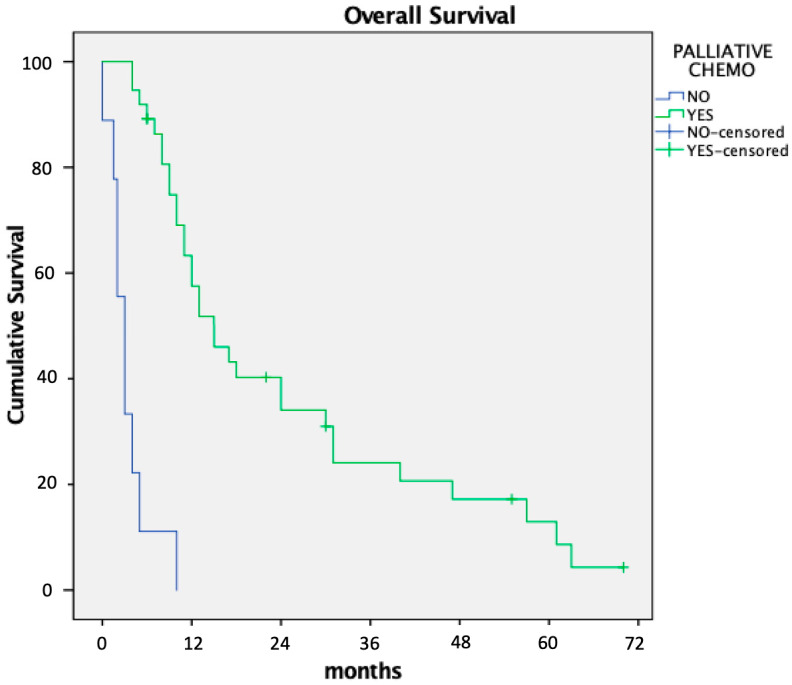
Influence on overall survival of postoperative palliative chemotherapy after nontherapeutic laparotomy.

**Table 1 cancers-17-01445-t001:** Main clinicopathological characteristics of the patients included.

Variable	n = 46
Age (median and range, in years)	60 (35–82)
<50	7 (15%)
50–70	31 (67%)
>70	8 (18%)
Tumor Origin (n and %)	
Ovarian	30 (65%)
Colon	8 (18%)
Endometrial	3 (6.5%)
Appendix (adenocarcinoma)	3 (6.5%)
Gastric	2 (4%)
Comorbidities (n and %)	
NO	22 (48%)
YES	24 (52%)
Previous Abdominal Surgery (n and %)	
NO	22 (48%)
YES	24 (52%)
ASA (n and %)	
II	27 (59%)
III	19 (41%)
CRS + HIPEC (n and %)	
Primary after neoadjuvant chemotherapy	28 (61%)
Recurrence	18 (39%)
Causes Of Nontherapeutic Laparotomy (n and %)	
Massive peritoneal carcinomatosis	28 (61%)
Massive small bowel involvement	13 (28%)
Frozen pelvis	4 (9%)
Porta hepatis infiltration	1 (2%)

**Table 2 cancers-17-01445-t002:** Postoperative data.

Variable	n = 46
Postoperative Morbidity (n and %)	
NO	36 (78%)
YES	10 (22%)
Postoperative Morbidity Grade (n and %)	
II	3 (7%)
III	2 (4%)
IV	3 (7%)
V	2 (4%)
Type of Complications (n and %)	
Paralytic ileus	3 (7%)
Wound seroma	3 (7%)
Ascites with paracentesis	2 (4%)
Acute kidney injury with multiorgan failure	2 (2%)
Reasmision (n and %)	
NO	43 (93%)
YES	3 (7%)
Readmitance Causes (n and %)	
Abdominal pain	1 (2%)
Paralytic ileus	1 (2%)
Small bowell obstruction	1 (2%)
Surgery At Readmitance (n and %)	
NO	2 (4%)
YES	1 (2%)

## Data Availability

To access the data, please contact the corresponding author: cascalescirugia@gmail.com.

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
