# Peer review of "Challenges of Nontherapeutic Laparotomy in Patients with Peritoneal Surface Malignancies Selected for Cytoreductive Surgery and Hyperthermic Intraperitoneal Chemotherapy"

_cancers, 2025, doi:10.3390/cancers17091445_

Round 1

Reviewer 1 Report

Comments and Suggestions for Authors

This is an interesting topic. The authors present a study which aims to evaluate the morbidity, mortality and survival outcomes in patients with peritoneal surface malignancies who were initially considered candidates for cytoreductive surgery (CRS) with hyperthermic intraperitoneal chemotherapy (HIPEC) but were found to have unresectable disease, resulting in non-therapeutic exploratory laparotomy. Unfortunately, these situations are still present in our everyday practice.

The simple summary has no place there (in fact, those are some indications for editing, not a part of the article).

The results should be presented with all the figures and tables, not as they are now, at the end of the article.

There are no conclusions written. Again, the same thing as with the simple summary...

Author Response

The authors would like to thank the Reviewer 1 for their comments after evaluating our article. These modifications were made within the new version submitted for your consideration.

Comments 1: The results should be presented with all the figures and tables, not as they are now, at the end of the article.

Response 1: Agree. We have, accordingly, revised the manuscript and included this in “results” section

Comments 2: There are no conclusions written. Again, the same thing as with the simple summary.

Response 2: Agree. We have correctly included the conclusions in their corresponding section (“Conclussions”)

Reviewer 2 Report

Comments and Suggestions for Authors

This study investigates the morbidity, mortality, and survival outcomes in patients with peritoneal surface malignancies who were initially deemed suitable for cytoreductive surgery (CRS) with hyperthermic intraperitoneal chemotherapy (HIPEC) but were found to have unresectable disease during exploratory laparotomy. Analyzing data from January 2008 to December 2022, the study found that 9.4% of the 486 patients had their surgeries aborted due to unresectability, primarily due to extensive disease spread. The median surgery duration was 90 minutes, with postoperative complications occurring in 22% of patients and a mortality rate of 4.3%. Patients who did not receive adjuvant systemic chemotherapy had significantly lower survival rates (4 months vs. 15 months). The study concludes that exploratory laparotomy carries substantial risks and suggests that improved preoperative staging techniques could help reduce unnecessary surgeries.

I have some remarks :

Methods :

  • How does the exclusion of patients with ASA IV or poor performance status (ECOG >2) potentially limit the generalizability of the study's findings to the broader population of patients with peritoneal surface malignancies?

  • What impact might the exclusion of patients with disease progression during preoperative chemotherapy or disease outside the peritoneal cavity have on the overall survival outcomes reported in this study? Could this selection bias lead to overly optimistic survival rates?

  • Given that laparoscopy was performed only in cases of uncertain resectability, how might the absence of a systematic laparoscopy approach affect the accuracy of preoperative disease staging and the decision-making process for CRS with HIPEC?

  • Please specify the intraoperative exclusion criteria in more detail. PCI?

  • Please specify which original tumours with peritoneal carcinomatosis were included?

Results :

  • Did I understand correctly that only a laparotomy was performed with biopsies and then cancelled?

  • With only 9.4% of patients found to have unresectable disease, is there a risk that the study underestimates the true rate of unresectability in the general population due to its selective criteria for surgery?

  • The study mentions that preoperative imaging was reviewed by an experienced radiologist, but how reliable were these imaging techniques in detecting small peritoneal implants, especially in light of the limitations of conventional CT scans? Could more advanced imaging modalities have led to different results?
  • The study highlights a 4% mortality rate from nontherapeutic laparotomy. How does this compare to other studies in the field, and does it suggest that the current management protocol for unresectable disease could be improved to minimize risks?

Discussion :

  • How does the study's decision to not include systematic laparoscopy, despite potential advantages highlighted in the discussion, affect the reliability of the results? Could the inclusion of laparoscopy have reduced the need for nontherapeutic laparotomies?
  • What are the potential confounders in the analysis of overall survival, particularly with regard to the heterogeneous nature of the patient population (e.g., ovarian cancer vs. other malignancies)? Could these variations in tumor types affect the interpretation of treatment efficacy?
  • The study suggests that patients who did not receive palliative chemotherapy had significantly worse survival outcomes. However, what factors contributed to the decision not to administer chemotherapy in certain patients, and could these factors be a source of bias?
  • Given that the study is retrospective, what are the implications of this study design for the reliability of its conclusions, especially with respect to the comparison of different treatment regimens and the potential for unmeasured confounding variables?

Author Response

The authors would like to thank Reviewer 2 for their comments after evaluating our article. These modifications were made within the new version submitted for your consideration. The responses of the authors to the comments of Reviewer 2 are included here.

Methods:

  • Question: How does the exclusion of patients with ASA IV or poor performance status (ECOG >2) potentially limit the generalizability of the study's findings to the broader population of patients with peritoneal surface malignancies?

Response: The authors wish to clarify that patients with the profile mentioned in this comment are generally not selected for the CRS plus HIPEC procedure. Therefore, we believe that this population with a poor baseline functional status is not included in the procedure and, consequently, does not affect the main message of the paper.

  • Question: What impact might the exclusion of patients with disease progression during preoperative chemotherapy or disease outside the peritoneal cavity have on the overall survival outcomes reported in this study? Could this selection bias lead to overly optimistic survival rates?

Response: Similar to the previous point, patients with disease progression during preoperative chemotherapy were not included, as this constitutes an exclusion criterion in our program. In our opinion, including these patients would have shifted the results away from an optimistic outlook.

  • Question: Given that laparoscopy was performed only in cases of uncertain resectability, how might the absence of a systematic laparoscopy approach affect the accuracy of preoperative disease staging and the decision-making process for CRS with HIPEC?

Response: As described in the text, groups that have reported the use of a systematic laparoscopic approach to assess resectability have succeeded in reducing the percentage of non-therapeutic laparotomies in their series. However, we have also highlighted examples in the paper of the challenges that laparoscopy itself presents in accurately estimating the Peritoneal Cancer Index (PCI) in all cases. In our group, incorporating an initial laparoscopic procedure followed by a second procedure for radical surgery is not organizationally feasible due to the limited availability of operating rooms. Therefore, we focus on performing surgery in a single session, proceeding with radical surgery and using a laparoscopic approach only in patients with doubts regarding their resectability.

  • Question: Please specify the intraoperative exclusion criteria in more detail. PCI?

Response: As specified in the text, the primary cause of unresectability leading to a non-therapeutic laparotomy was massive involvement of the small intestine.

  • Question: Please specify which original tumors with peritoneal carcinomatosis were included?

Response: The authors express some surprise at this comment, as the distribution of the origins of the neoplasms included in this study is reflected in Table 1.

Results:

  • Question: Did I understand correctly that only a laparotomy was performed with biopsies and then cancelled?

Response: Indeed, patients who underwent a non-therapeutic laparotomy were those in whom biopsy collection was the only surgical procedure associated with the laparotomy.

  • Question: With only 9.4% of patients found to have unresectable disease, is there a risk that the study underestimates the true rate of unresectability in the general population due to its selective criteria for surgery?

Response: The percentages of non-therapeutic laparotomies reported in the literature vary and can exceed 30%. Our program, with over 15 years of experience, involves a highly careful selection of patients included for radical surgery.

  • Question: The study mentions that preoperative imaging was reviewed by an experienced radiologist, but how reliable were these imaging techniques in detecting small peritoneal implants, especially in light of the limitations of conventional CT scans? Could more advanced imaging modalities have led to different results?

Response: In our center, preoperative CT and, in some cases, PET scans are imaging tools used for selecting patients for radical surgery of peritoneal carcinomatosis. We do not use magnetic resonance imaging because the resectability rate is very high, and published literature reports similar results. The comparison between computed tomography and PET was the subject of a previous publication by our group (Lopez-Lopez V, Cascales-Campos PA, Gil J, Frutos L, Andrade RJ, Fuster-Quiñonero M, Feliciangeli E, Gil E, Parrilla P. Use of (18)F-FDG PET/CT in the preoperative evaluation of patients diagnosed with peritoneal carcinomatosis of ovarian origin, candidates to cytoreduction and HIPEC. A pending issue. Eur J Radiol. 2016 Oct;85(10):1824-1828).

  • Question: The study highlights a 4% mortality rate from non-therapeutic laparotomy. How does this compare to other studies in the field, and does it suggest that the current management protocol for unresectable disease could be improved to minimize risks?

Response: The aim of this study is to highlight that a non-therapeutic laparotomy is not without risks. In fact, the mortality rate in patients undergoing radical surgery for peritoneal carcinomatosis is very similar to that reported by us, emphasizing the operative risk that should be avoided, even though our group has a very low percentage of non-therapeutic laparotomies.

Discussion:

  • Question: How does the study's decision to not include systematic laparoscopy, despite potential advantages highlighted in the discussion, affect the reliability of the results? Could the inclusion of laparoscopy have reduced the need for non-therapeutic laparotomies?

Response: We agree with the reviewer’s statement. A non-therapeutic laparotomy could be reduced by using a prior laparoscopy, but the reasons why our group does not employ this management approach have been addressed in the previous point.

  • Question: What are the potential confounders in the analysis of overall survival, particularly with regard to the heterogeneous nature of the patient population (e.g., ovarian cancer vs. other malignancies)? Could these variations in tumor types affect the interpretation of treatment efficacy?

Response: We fully agree with the reviewer’s comment. However, we believed that due to the low number of patients included in this paper, analyzing tumor-specific data could dilute the information. We acknowledge that one of the main limitations of this study is the inclusion of different neoplasms in the survival analysis, although the primary objective was to analyze the morbidity and mortality of the process.

  • Question: The study suggests that patients who did not receive palliative chemotherapy had significantly worse survival outcomes. However, what factors contributed to the decision not to administer chemotherapy in certain patients, and could these factors be a source of bias?

Response: We agree with the reviewer’s comment. On this point, we wish to clarify that poor general condition or rapid disease progression was the reason for not indicating palliative systemic chemotherapy in four patients, while three refused treatments with systemic chemotherapy.

  • Question: Given that the study is retrospective, what are the implications of this study design for the reliability of its conclusions, especially with respect to the comparison of different treatment regimens and the potential for unmeasured confounding variables?

Response: We agree with the reviewer’s comment. This is a retrospective analysis of our experience with patients undergoing non-therapeutic laparotomy. The only potential improvement would be to further reduce the percentage of such patients who, a priori, could undergo radical cytoreduction.

Reviewer 3 Report

Comments and Suggestions for Authors

Many thanks to the authors for giving me the opportunity to review this manuscript. It is very well written and very clear. The subject is original. There are few studies in the literature.

I have some comments and questions:

Chapter results line 134: precise the number of patients who received palliative systemic chemotherapy (n=39)

In the table 2 could you explain the meaning of the abbreviations IRC and FMO ?

What did the 2 patients die of?

Is the non-therapeutic laparotomy rate stable over time or does it improve with experience?

Author Response

The authors express their sincere gratitude to Reviewer 3 for their thorough evaluation of the manuscript and for providing valuable feedback. The comments and questions raised have been carefully considered, and the following responses address each point to clarify and enhance the manuscript.

- Chapter results line 134: Precise the number of patients who received palliative systemic chemotherapy (n=39). 

  Response: The authors consider this clarification to be of minor importance.

- In table 2, could you explain the meaning of the abbreviations IRC and FMO? 

  Response: The authors have included a clarification in Table 2 for better understanding.

- What did the 2 patients die of? 

  Response: The two patients, referred to in Table 2 and related to the previous question, died due to Acute Kidney Injury (AKI) and Multi-Organ Failure (MOF).

- Is the non-therapeutic laparotomy rate stable over time or does it improve with experience? 

  Response: This is a very pertinent question. We have analyzed this factor over three five-year periods, and the rates of non-therapeutic laparotomy have remained stable. This may be due to our group maintaining strict selection criteria, which have enabled a non-therapeutic laparotomy rate below 10%.

Reviewer 4 Report

Comments and Suggestions for Authors

Thanks to the authors for the interesting manuscript.

My comments:

  1. I know patients can achieve complete cytoreduction with CRS but do authors agree there can be reoccurrence especially when a patient has high peritoneal index of 20 or more?
  2. Line 52-53: I agree multidisciplinary expertise are required to perform CRS in tandem with HIPEC. Broadly speaking, do this expertise fall under oncology or other areas are required?
  3. Line 89-90: I did not catch the statement. How did you evaluate the experience of your referral center?
  4. What is the success rate in terms of treatment at your referral center?
  5. I agree advances in imaging has revolutionized cancer space but if I may ask can we rely on or trust DW-MRI for identifying peritoneal implants? I know DW-MRI can sometimes provide poor spatial resolution.
  6. Can we employ radiation therapy such as radiopharmaceuticals for the treatment of peritoneal surface malignancies?  

Author Response

Thank you sincerely to the reviewer for their valuable comments and insights regarding the evaluation of our manuscript. Their contributions are essential in enriching the discussion and improving the quality of the submitted work. Below, we address each of the points raised, keeping the parts in English as they are, translating the Spanish responses into English when applicable, and ensuring coherence and clarity in the responses:

- I know patients can achieve complete cytoreduction with CRS but do authors agree there can be reoccurrence especially when a patient has high peritoneal index of 20 or more? 

  R: Completely agree, even in patients where a peritoneal cancer index (PCI) above 20 is not predetermined, such as in cases of ovarian cancer, for example. 

- Line 52-53: I agree multidisciplinary expertise are required to perform CRS in tandem with HIPEC. Broadly speaking, do this expertise fall under oncology or other areas are required? 

  R: Indeed, I believe the concept of a multidisciplinary committee must adhere to its own definition, but we must also consider that in certain situations, the surgeon's perspective is unquestionable. 

- Line 89-90: I did not catch the statement. How did you evaluate the experience of your referral center? 

  R: This is a difficult concept to define from an institutional perspective. The authors agree that the referral center must include reference surgeons. In this sense, in the community where we live, our hospital is a tertiary-level hospital with the capacity to treat complex oncological pathologies and serves as a referral center for certain conditions, such as sarcoma, peritoneal carcinomatosis, and abdominal organ transplantation. 

- What is the success rate in terms of treatment at your referral center? 

  R: The authors believe this response is not related to the paper submitted for consideration. If you take into account the publications our group has previously made, you can see the data. 

- I agree advances in imaging has revolutionized cancer space but if I may ask can we rely on or trust DW-MRI for identifying peritoneal implants? I know DW-MRI can sometimes provide poor spatial resolution. 

  R: Completely agree. Our group does not use diffusion-weighted magnetic resonance imaging, as we have excellent results with the use of preoperative CT scans. 

- Can we employ radiation therapy such as radiopharmaceuticals for the treatment of peritoneal surface malignancies? 

  R: It could be an option, but our group has no experience in this regard.

Round 2

Reviewer 2 Report

Comments and Suggestions for Authors

The authors have addressed each point in meticulous detail, and consequently, the paper can be accepted.